# DEBIASING VISON-LANGUAGE MODELS WITH TEXT-ONLY TRAINING

## ABSTRACT

Pre-trained vision-language models (VLMs), such as CLIP, have exhibited remarkable performance across various downstream tasks by aligning text and images in a unified embedding space. However, due to the imbalanced distribution of pre-trained datasets, CLIP suffers from the bias problem in real-world applications. Existing debiasing methods struggle to obtain sufficient image samples for minority groups and incur high costs for group labeling. To address the limitations, we propose a **T**ext-**O**nly **D**ebiasing framework called **TOD**, leveraging a text-as-image training paradigm to mitigate visual biases. Specifically, this approach repurposes the text encoder to function as an image encoder, thereby eliminating the need for image data. Simultaneously, it utilizes a large language model (LLM) to generate a balanced text dataset, which is then used for prompt tuning. However, we observed that the model overfits to the text modality because label names, serving as supervision signals, appear explicitly in the texts. To address this issue, we further introduce a Multi-Target Prediction (MTP) task that motivates the model to focus on complex contexts and distinguish between target and biased information. Extensive experiments on the Waterbirds and CelebA datasets show that our method significantly improves group robustness, achieving state-of-the-art results among image-free methods and even competitive performance compared to image-supervised methods. Furthermore, the proposed method can be adapted to challenging scenarios with multiple or unknown bias attributes, demonstrating its strong generalization and robustness.

## 1 INTRODUCTION

In recent years, pre-trained VLMs have made significant progress. Benefitting from large amounts of image-text data, the VLMs, such as CLIP (Radford et al., 2021) and ALIGN (Jia et al., 2021), learn rich visual and textual representations. Specifically, these models construct a joint visual-language embedding space where semantically relevant images and text are encoded as close features. This aligned embedding space enables CLIP to excel in various downstream tasks, such as zero-shot classification (Guo et al., 2023b), image-text retrieval (Baldrati et al., 2022) and image captioning (Mokady et al., 2021).

While CLIP demonstrate impressive capabilities, it inherits biases or spurious correlations from the inappropriate pre-training data. This behavior leads to poor group robustness in downstream tasks—the model performs significantly worse on some groups where the correlations do not hold. The lack of group robustness not only weakens the generalization ability of the model but also has the potential to exacerbate existing social biases such as racism or gender discrimination. Therefore, addressing the bias and robustness challenges in CLIP models has become a critical issue.

Many works aim to improve group robustness and have been applied to vision-language models. Previous methods typically retrain or fine-tune the model on downstream visual tasks with group annotations (Sagawa et al., 2019; Byrd & Lipton, 2019; Idrissi et al., 2022; Kirichenko et al., 2023). Other works have explored debiasing of CLIP without group labels, such as deriving them from zero-shot predictions(Liu et al., 2021; Nam et al., 2020), to lower annotation costs. However, acquiring sufficient images for minority group (such as waterbirds on land) can be expensive or even infeasible, while the training process on image may be computationally intensive. Therefore, recent works (Chuang et al., 2023) have proposed training-free debiasing methods, but achieved inferior

performance compared to those using image supervision. To address these limitations, we propose a **T**ext-**O**nly **D**ebiasing framework called **TOD**, which leverages CLIP's ability to align across both image and text modalities, exploiting textual data to mitigate visual biases.

To the best of our knowledge, this is the first work to leverage a text-as-image (TaI) training paradigm (Guo et al., 2023a) to mitigate visual biases. TOD comprises two stages: balanced text data generation and text-only training. Given a set of categories and biased attributes, we first employ GPT-4o to generate textual descriptions that contain the target and attribute names, yielding a distribution-balanced training dataset. In this way, group labels can be directly derived from target and attribute names. Then, the text descriptions serve as alternatives to images in prompt tuning, and the learned prompt can classify images during inference. Compared to images, our approach is more cost-effective and scalable by utilizing the capabilities of LLMs to automatically generate balanced text data and group labels.

While this method reduces the reliance on costly image data, we observe that text-only training can lead to overfitting to the text modality, exhibiting poor generalization when transferred to classify images as shown in Figure 2. This is primarily because the classification in text-only training can be easily made by matching the class names in prompts with similar words in text descriptions, which is less influenced by bias information compared to image-text matching. To address this issue, inspired by human visual perception which processes multiple objects in parallel (Brandman & Peelen, 2017; de Wit et al., 2011), we intoduce a Multi-Target Prediction (MTP) task. This task involves simultaneously predicting target attributes and bias attributes within a single prompt. By compelling the model to actively focus on different attribute words or image regions, MTP increases task difficulty, thereby mitigating overfitting. Unlike previous works that remove bias information from representations (Chuang et al., 2023; Berg et al., 2022), our method guides the model to distinguish between the concepts of target attributes and biased attributes, achieving a more robust debiasing effect. Experiments on the Waterbids and CelebA datasets show that our method significantly outperforms existent image-free methods and even surpasses image-supervised methods. Moreover, the approach can be naturally adapted to more challenging scenarios, delivering robust performance in the presence of multiple bias attributes or when the bias attributes are unknown.

Our main contributions are summarized as follows:

1. We propose a novel **T**ext-**O**nly **D**ebiasing framework **(TOD)**, which mitigates bias in CLIP by utilizing LLMs to generate balanced text datasets, circumventing the need for visual data.

2. We introduce an effective solution for modal overfitting in text-only training: Multi-Target Prediction (MTP) with prompt tuning. This significantly enhances the model's debiasing performance when applied to image data.

3. Our method significantly improve robustness group robustness on the Waterbirds and CelebA datasets, achieving performance comparable to state-of-the-art image-supervised methods. Besides, TOD can be seamlessly extended to scenarios with multiple bias attributes and unknown bias attributes, demonstrating its strong generalization and robustness.

## 2 RELATED WORK

**Group Robustness of Vison Models.** Many works use supervision of group labels in training to improve the group robustness of vision models. This includes strategies such as minimizing the loss of the worst group for robustness optimization (Sagawa et al., 2019), importance reweighting (Byrd & Lipton, 2019; Shimodaira, 2000) or subsampling to balance the samples of major and minor groups (Idrissi et al., 2022; Kirichenko et al., 2023). All of these methods require group labels for the entire training set, which can be extremely expensive. Recent research has circumvented the need of group annotations during training, instead relies only on a small validation set with group labels for hyperparameter selection. A common approach is to first train a model to infer the group labels of the samples and then train a second robust model. For example, BPA (Seo et al., 2022) automatically infers group labels and performs robust training by clustering sample points. JTT (Liu et al., 2021) and LfF (Nam et al., 2020) reweight difficult or misclassified samples. CNC (Zhang et al., 2022) learns similar representations for samples from different groups of the same class by

contrastive loss. However, all of these methods rely on the downstream training set and often train the visual model twice, which is costly when applied to large foundation models.

**Debiasing Vison-Language Models.** Recently, the bias issue in vision-language models (e.g., CLIP) has garnered increasing attention (Hall et al., 2023; Agarwal et al., 2021; Slyman et al., 2023). Considering the huge number of parameters in VLMs, many studies have focused on eliminating spurious correlations to the bias attribute without updating the entire network weights. For example, Berg et al. (2022) mitigates bias in CLIP via adversarial prompt tuning, Zhang & Ré (2022) uses contrastive learning to train an adapter (a 2-layer MLP) in the representation space. Meanwhile, Seth et al. (2023) introduces an additional residual module to separate protected attribute information from the visual representation. Dehdashtian et al. (2024) employs a kernel approach to debias CLIP, removing bias without the need for attribute annotations. However, these methods require a downstream image training set. Our approach requires only the target category and bias information without any training images. The closest related works aim to eliminate biases in vision-language models in a zero-shot manner. Chuang et al. (2023) defines the directions of spurious correlation with biased prompts and projects them out from the text embedding. However, these methods have a large gap in performance with the trained methods, while our method has a competitive or even surpassing effect compared to the methods trained on real data.

**Text as Image Training.** The CLIP encoder maps images and text into a shared embedding space where semantically similar images and text are aligned with each other. This allows the use of features from another modality in place of the original modality during the training phase. Since text data is more accessible and contains richer vision concepts which facilitate model to learn generalized representations, many works use text as a substitute for images in training. For example, Nukrai et al. (2022); Li et al. (2023); Fei et al. (2023) train a decoder to reconstruct the masked text and generate image captions by replacing the text input with images in the inference phase. Guo et al. (2023a); Zhu et al. (2023) exploited the inherent property of textual descriptions which contain semantic dependencies and spatial relationships between objects, to perform text-only training for multi-labeled image recognition. Additionally, several works have employed text-as-image training for tasks such as composed image retrieval (Gu et al., 2024), visual question answering (VQA) (Gu et al., 2023), and visual entailment (Gu et al., 2023). Our work further explores the application scenarios of text-as-image training by applying it to debiasing tasks for the first time.

## 3 PRELIMINARIES

**Notation.** In this work, we consider group robustness as the measure of bias, following the mainstream practice (Sagawa et al., 2019; Creager et al., 2021; Liu et al., 2021; Kirichenko et al., 2023). We denote the input image as $x \in X$, the target category as $y \in Y$, and the bias attribute category as $b \in B$. Take the Waterbirds dataset (Sagawa et al., 2019) as an example, where the target task is to recognize birds with category $y \in \{landbird, waterbird\}$, and the bias attribute is the background $b \in \{land, water\}$. The combination of the target category and the bias attribute category divides the dataset into groups, denoted as $g \in G$, with the group labeling denoted as $G = Y \times B$. When the attribute s affects the prediction of y but lacks a causal relationship, it is considered as bias correlation. For example, in the Waterbirds dataset, 95% of the samples with $y = waterbird$ have the bias attribute $b = water$. As a result, models trained on this dataset may rely heavily on the background (water) to predict the category (waterbirds), leading to incorrect predictions on the minority group $g = (water, landbird)$. To measure the debiasing effect, we use the worst group accuracy (WG) and the gap between the average accuracy and the worst group accuracy (Gap) as our evaluation metrics.

**Revisiting CLIP.** The core idea of CLIP (Radford et al., 2021) is to train a model capable of embedding text and images into a single embedding space, thus realizing cross-modal information understanding and processing. CLIP consists of an image encoder $\Phi_I$ and a text encoder $\Phi_T$, which convert high-dimensional image $X$ and text sequences $T$, respectively, into embedding vectors in a shared low-dimensional space. Specifically, this objective aims to maximize the cosine similarity $< z_I, z_T > = \frac{z_I \cdot z_T}{\|z_I\| \cdot \|z_T\|}$ between the matched image-text pairs, while minimizing the similarity between the non-matched pairs. Here $z_I$ and $z_T$ represent the output embedding vectors of image encoder $\Phi_I$ and text encoder $\Phi_T$, respectively. By training on a large dataset containing 400 million

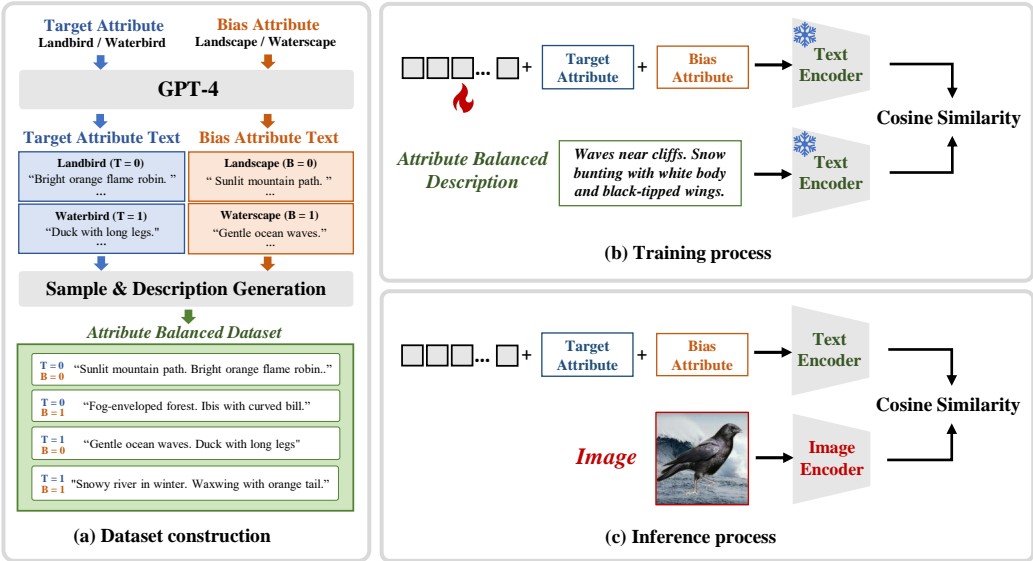

Figure 1: **Overview of Text-Only Debiasing (TOD) framework.** (a) Construction of a balanced text dataset using GPT-4o. First, we generate a set of text descriptions for the target attributes and false attributes separately. Then, we randomly sample from these sets and concatenate them to create text descriptions with group labels. Both training and inference process is based on multi-target prediction, which simultaneously predicts target attributes and bias attributes. (b) During training, we use using two identical, frozen text encoders from pre-trained CLIP that separately encode the text descriptions and class prompts. The model is optimized through prompt tuning. (c) During inferencing, we replace the input from text descriptions to images, and take the target attribute from the group with highest logits as the final prediction.

image-text pairs, CLIP successfully aligns text embeddings and image embeddings semantically in the shared embedding space. For image-text pairs with similar semantics, the vison embeddings generated by the image encoder will be close to the text embeddings generated by the text encoder. Therefore, texts inputs can serve as substitutes of images for training in downstream tasks.

## 4 METHOD

We propose a novel debiasing method for CLIP: multi-target prediction based on text as image training, with the framework shown in Figure 1. Initially, we generate a balanced text dataset for the downstream classification task using LLM in Section 4.1. During the training phase, we fine-tuning the model on the text dataset in Section 4.2 *(Training Phase)*. Particularly, we employ a multi-target prediction task that simultaneously predicts target and bias categories to mitigate overfitting and imitate human visual perception. During the inference phase, we perform multi-target prediction on the image input and select the combination of the target and bias categories with the highest logits, where the target category serves as the final classification result in Section 4.2 *(Inference Phase)*.

### 4.1 TEXT DATASET GENERATION

**Attribute Description Generation.** The first step of our method is to determine the attributes contained within the dataset. Taking the Waterbird dataset as an example, we defined two key attributes: the target attribute (bird) and the bias attribute (background). The target attribute is divided into two categories: waterbird and landbird, while the bias attribute is divided into landscape and waterscape. We applied the GPT-4o to generate descriptions of these attributes. The specific procedure was as follows: we instructed the model to generate 1,000 short image descriptions of waterbirds, and the instruction used was, "*please help me to generate 100 very short images captions of waterbird. Be careful to describe only the appearance of the bird, not the background or environment.*" This

process was repeated 10 times. We generated descriptions for each attribute class, including 1,000 descriptions each for waterbird, landbird, landscape, and waterscape.

**Attribute Combination with Automatic Annotation.** After generating independent attribute descriptions, we compose the attributes to construct a complete image description. Unlike the coupling between features in images, feature composition can be achieved in text by just concatenation. Specifically, we randomly select a bird description from the waterbird or landbird description sets and a background description from the landscape or waterscape description sets. These two parts are then concatenated to form a complete image description that included both the foreground and the background. To ensure a balanced training dataset, we generated an equal number of descriptions for each attribute composition.

This design automates the data annotation process since the source of each attribute description is predefined with the corresponding label. When the bird description and background description come from the waterbird description set and landscape description set respectively, we can directly assign the target label T=1 (for waterbird) and the bias label B=0 (for landscape). Meanwhile, the group label G is defined as the composition of attributes, e.g., G=1×0. The high annotation cost of traditional image datasets is effectively avoided.

Following the steps, we generated balanced text datasets containing 10,000 descriptions each for the Waterbirds and CelebA datasets. Dataset construction details for CelebA are provided in the Appendix A.2.

## 4.2 MULTI-TARGET PREDICTION

**Training Phase.** We use prompt tuning on multi-target prediction to predict both target and bias attributes. A prompt is defined as

$$t_{i,j}^T = [v_1, v_2, v_3, ..., target_i \oplus bias_j], \tag{1}$$

where $i \in \{1, 2, 3, ..., C_T\}$ is the target class index, $j \in \{1, 2, ..., C_b\}$ is the bias class index, $target_i$ and $bias_j$ are word embeddings of the $i^{th}$ target class name and the $j^{th}$ bias class name, and $\oplus$ denotes a concatenator, usually a comma or other designed word. For $k \in \{1, 2, ..., M\}$, $v_k$ is a learnable word embedding whose dimension is the same as the dimension of normal word embeddings in the vocabulary. $M$ is a hyperparameter specifying the number of learnable word embeddings. By forwarding a prompt to the text encoder, we can obtain a classification weight vector representing the target class $i$ and the bias class $j$. Different from image-based prompt tuning, we input text descriptions $x$ into a same text encoder $\Phi_T$ to get text embeddings for classification.

Following the previous prompt tuning methods (Zhou et al., 2022), learnable prompts are optimized by maximizing the probability of classifying each text description into the ground-truth composition of the target and bias classes. The prediction probability is computed as

$$p(y = i, b = j|x) = \frac{\exp(< \Phi_T(t_{i,j}^T), \Phi_T(x)) > \tau)}{\sum_{i=1}^{C_T} \sum_{j=1}^{C_B} \exp(< \Phi_T(t_{i,j}^T), \Phi_T(x) > /\tau)}. \tag{2}$$

Since the value of cosine similarities between text descriptions and class prompts are not evenly distributed on either side of 0, directly constraining the softmax probability makes the optimization more difficult in this case. Therefore, we use ranking loss (Gong et al., 2013) instead of cross-entropy loss for training to minimize the multi-target classification loss. Given $P_{i,j} = P(y = i, b = j|x)$, the loss function is formulated as follows:

$$\mathcal{L} = \sum_{(i,j) \neq (y^*, b^*)} \max(0, \gamma - P_{y^*, b^*} + P_{i,j}), \tag{3}$$

where $y^*$ and $b^*$ are the target and bias class labels, and $\gamma$ denotes the margin that controls how much the similarity score with the positive class is higher than with the negative class. During training, we freeze the two text encoders to minimize the loss by optimizing the learnable word embeddings $v_k, k \in \{1, 2, ..., M\}$.

**Inference phase.** Due to text and images sharing a unified embedding space, the optimized prompt is directly applicable to the image modality. Input a test image $x$ into the image encoder $\Phi_I$ to obtain an image embedding, and calculate the cosine similarity with the prompt embeddings. The prediction probability is calculated as follows

$$p(y = i, b = j|x) = \frac{\exp(<\Phi_T(t_{i,j}^T), \Phi_I(x)) > \tau)}{\sum_{i=1}^{C_T} \sum_{j=1}^{C_B} \exp(<\Phi_T(t_{i,j}^T), \Phi_I(x) > /\tau)}. \tag{4}$$

sSo the target class prediction can be directly computed as

$$\hat{y} = \arg \max_i \max_j p(y = i, b = j|x), \tag{5}$$

i.e., the target class in the target- bias attribute composition with the highest probability.

## 5 EXPERIMENT

### 5.1 SETUP

**Evaluated Methods.** As baselines for comparison, we consider the methods based on image training sets, including ERM Linear (Kumar et al., 2022; Radford et al., 2021), ERM Adapter (Gao et al., 2024), WiSE-FT (Wortsman et al., 2022), DFR (Kirichenko et al., 2023), Contrastive Adapter (Zhang & Ré, 2022), FairerCLIP (Dehdashtian et al., 2024), and DPS+RNS (You et al., 2024). We also compare with image-free methods including Orth-Proj and Orth-Cali (Chuang et al., 2023).

**Dataset.** We conduct experiments on commonly used datasets for bias and fairness assessment.

- Waterbirds (Sagawa et al., 2019) combines the Caltech-UCSD Birds-200-2011 (CUB) dataset (Wah et al., 2011) and the Places dataset (Zhou et al., 2017), placing images of birds on different backgrounds to simulate different biases. The task is to classify foreground birds (waterbird or landbird) and the background (land or water landscape) is a bias attribute. Statistically, the minimum group in the training set contains only 53 images.

- CelebA (Liu et al., 2015) contains over 200,000 portraits of celebrities, each annotated with 40 binary attribute labels such as hair color, gender and age. The task on this dataset is to classify hair color (blond or non-blond) and gender (male or female) is a bias attribute. Statistically, the women examples is more than 94% in the blond hair of training set. Additionally, we consider more complex scenarios including multiple bias attributes and unknown bias attributes on the CelebA.

**Implementation Details.** We evaluate the performance of our method and baselines using two CLIP backbones: ResNet-50 and ViT-L/14. Typically, methods without group label supervision (Zhang & Ré, 2022; You et al., 2024) still utilize group labels in the validation set. Following this setup, we use the worst group accuracy of the image validation set to select the stop epoch and hyperparameters. In scenarios where the bias attribute is unknown, we select the stop epoch and model hyperparameters in terms of the average accuracy.More detailed settings are shown in the Appendix A.1 .

**Evaluation Protocol.** For performance evaluation, we use three metrics: 1) Average accuracy (Avg.), 2) Worst-group accuracy (WG), i.e., the lowest accuracy of all subgroups, and 3) Gap, which is the difference between average and worst-group accuracy.

### 5.2 RESULTS IN SIMPLE DEBIASING SCENARIOS

We present our main results on standard benchmark, including CelebA and Waterbird datasets. Furthermore, we investigate the generalizability of our method by evaluating it on more bias attributes.

**Standard Benchmarks.** Table 1 presents a comparison of our algorithm with existing methods on the Waterbirds and CelebA datasets. The results indicate that TOD consistently achieves the best worst-group accuracy and the smallest robustness gaps compared with the state-of-the-art image-free

| Backbone | CLIP ViT-L/14 | | | | | | CLIP ResNet-50 | | | | | |
|---|---|---|---|---|---|---|---|---|---|---|---|---|
| | **Waterbird** | | | **CelebA** | | | **Waterbird** | | | **CelebA** | | |
| Method / Acc.(%) | WG (↑) | Avg (↑) | Gap (↓) | WG (↑) | Avg (↑) | Gap (↓) | WG (↑) | Avg (↑) | Gap (↓) | WG (↑) | Avg (↑) | Gap (↓) |
| *methods without image data* | | | | | | | | | | | | |
| ERM Linear | 65.9 | 97.6 | 31.7 | 28.3 | 94.7 | 66.4 | 7.9 | 93.5 | 85.6 | 11.9 | 94.7 | 82.8 |
| ERM Adapter | 78.4 | 97.8 | 19.4 | 36.7 | 94.2 | 57.5 | 60.8 | 96.0 | 35.2 | 36.1 | 94.2 | 58.1 |
| WiSE-FT | 65.9 | 97.6 | 31.7 | 80.0 | 87.4 | 7.4 | 49.8 | 91.0 | 41.2 | 85.6 | 88.6 | 3.0 |
| DFR (Sub) | 51.9 | 95.7 | 43.8 | 76.3 | 92.1 | 15.8 | 63.9 | 91.8 | 27.9 | 76.9 | 92.5 | 15.6 |
| DFR (Up) | 65.9 | 96.1 | 30.2 | 83.7 | 91.2 | 7.5 | 51.3 | 92.4 | 41.1 | 89.6 | 91.8 | 2.2 |
| Con-Adapter | 86.9 | 96.2 | 9.3 | 84.6 | 90.4 | 5.8 | 83.7 | 89.4 | 5.7 | 90.0* | 90.7 | 0.7* |
| FairerCLIP | 86.0 | 92.2 | **6.1** | 85.2 | 87.8 | **2.5** | 75.4 | 84.3 | 8.9 | 81.5 | 85.0 | 3.5 |
| DPS+RNS | **88.2*** | 96.8 | 8.6 | 84.8 | 87.8 | 3.0 | 76.9 | 77.6 | **0.7*** | 73.7 | 81.1 | 7.4 |
| *methods without image data* | | | | | | | | | | | | |
| Zero-shot | 45.3 | 84.4 | 39.1 | 72.8 | 87.6 | 14.9 | 39.6 | 77.3 | 37.7 | 75.9 | 82.3 | 6.4 |
| Orth-Proj | 61.4 | 86.4 | 25.0 | 71.1 | 87.0 | 15.9 | 48.1 | 83.6 | 35.4 | 61.4 | 86.4 | 25.0 |
| Orth-Cali | 68.8 | 84.5 | 15.7 | 76.1 | 86.2 | 10.1 | 74.0 | 78.7 | 4.7 | 82.2 | 84.4 | 2.2 |
| **TOD(Ours)** | **87.8** | 88.8 | **1.0*** | **85.3*** | 86.5 | **1.2*** | **84.0*** | 85.3 | **1.3** | 86.4 | 88.3 | **1.8** |

Table 1: Results on improving group robustness of CLIP models. We use birds and background as the classification target and bias attribute for the Waterbirds dataset, and hair color and gender as the target category and bias attribute for the CelebA dataset. For each CLIP's backbone, the first block of the table contains methods that require image data for training and the numbers are taken from Zhang & Ré (2022), and the second block of the table contains methods without image data and the numbers are taken from Chuang et al. (2023). The best worst-group accuracy (WG) and robustness gaps (Gap) in one block **bolded**, and ∗ denotes the method with the best performance in 2 blocks; we show means over 3 seeds.

methods Orth-Proj and Orth-Cali. Remarkably, TOD achieves competitive worst-group accuracy and robustness gaps compared to the leading supervised methods (e.g. Con-Adapter and DPS+RNS) and significantly outperforms other supervised methods (e.g. ERM Adapter, DFR(Sub), DFR(Up) and FairerCLIP), without any image training data. In addition, we observe that most methods fail to outperform previous approaches consistently on all settings and metrics, while TOD exhibits balanced and robust performance across differet datasets, model architectures, and metrics, demonstrating its stability and generalizability.

**On Different Bias Attributes.** We conducted additional debiasing experiments on the CelebA dataset to assess the general applicability of our approach across different biases. Focusing on hair color as the target attribute, we examined several bias attributes, including chubby, wearing hat, and age, with results shown in Table 2. The initial zero-shot results indicate that the CLIP exhibits significant spurious correlations on these attributes. For comparison, we selected representative image-supervised approaches (Con-Adapter and FairerCLIP) and image-unsupervised approach (Orth-Proj and Orth-Cali). According to Table 2, TOD consistently improve the worst-group accuracy over zero-shot classification by 9.8 to 22.4 pp across different bias attributes, and achieved the smallest gap in both image-supervised and image-free methods, showing the potential to address debiasing issues for complex and diverse bias categories.

## 5.3 RESULTS IN MORE CHALLENGING SCENARIOS

Existing methods usually focus on single-bias scenarios. In addition, most methods require access to bias types for training or for model selection on the validation set. Here, we consider the more challenging tasks of (1) De-biasing multiple bias attributes simultaneously. (2) Bias attributes are unknown throughout the training and validation process.

**Debiasing Multiple Bias Attributes.** We use combinations of multiple bias attributes on CelebA with the goal of obtaining a model robust to each bias attribute. By attaching additional prediction targets in the prompt, our method can naturally extend to multi-bias situation. For example, the prompt could be initialized as "A photo of a celebrity with blond hair, male, young" for the gender and age bias. As shown in Table 3, our method achieves the best worst group accuracy on all settings with improvements of 8.0 pp to 11.3 pp in gender bias, 17.5 pp to 20.0 pp in age bias, and 5.0 pp to 7.1 pp in wavy hair bias.

**Debiasing Unknown Bias Attributes.** Existing methods typically require prior information on which attributes the model is biased with. Methods such as Constractive Adapter (Zhang & Ré,

| Bias Attribute | Method | WG (↑) | Avg (↑) | Gap (↓) |
|---|---|---|---|---|
| Chubby | zero-shot | 61.9 | 82.4 | 20.5 |
| | Contrastive Adapter | **81.0** | 90.0 | 9.0 |
| | FairerCLIP | 55.8 | 84.8 | 28.9 |
| | Orth-Proj | 61.9 | 83.3 | 21.4 |
| | Orth-Cali | 66.7 | 80.5 | 13.9 |
| | **TOD** | **81.0** | 83.8 | **2.9** |
| Wearing_Hat | zero-shot | 61.9 | 82.4 | 20.5 |
| | Contrastive Adapter | **84.6** | 88.8 | 4.2 |
| | FairerCLIP | 54.4 | 87.8 | 33.4 |
| | Orth-Proj | 73.6 | 80.5 | 7.0 |
| | Orth-Cali | 78.1 | 83.8 | 5.7 |
| | **TOD** | 84.3 | 86.4 | **2.1** |
| Age | zero-shot | 77.9 | 82.4 | 4.5 |
| | Contrastive Adapter | **87.7** | 90.4 | 2.6 |
| | FairerCLIP | 82.2 | 85.7 | 3.5 |
| | Orth-Proj | 74.8 | 83.3 | 8.5 |
| | Orth-Cali | 76.1 | 83.6 | 7.5 |
| | **TOD** | **87.7** | 88.3 | **0.6** |

Table 2: Debiasing other bias attributes. Evaluate the debiasing effect on three bias attributes (Chubby, Wearing Hat and Age) on the CelebA dataset. We run experiments on CLIP ResNet-50. **1st** / 2nd best results are in **bolded** / underline.

| Bias Attributes | Method | Worst group accuracy (↑) | | | |
|---|---|---|---|---|---|
| | | Gender bias | Age bias | Wavy Hair bias | Average |
| Gender | zero-shot | 75.9 | 77.9 | —— | 76.9 |
| Age | Orth-Proj | 73.6 | 72.6 | —— | 73.1 |
| | Orth-Cali | 69.1 | 69.6 | —— | 69.3 |
| | **TOD** | **87.2** | **87.9** | —— | **87.6** |
| Gender | zero-shot | 75.9 | —— | 78.7 | 77.3 |
| Wavy hair | Orth-Proj | 82.2 | —— | 83.4 | 82.8 |
| | Orth-Cali | 82.5 | —— | 83.2 | 82.9 |
| | **TOD** | **83.9** | —— | **83.7** | **83.8** |
| Gender | zero-shot | 75.9 | 77.9 | 78.7 | 77.5 |
| Age | Orth-Proj | 79.5 | 79.5 | 77.0 | 78.7 |
| Wavy hair | Orth-Cali | 75.5 | 84.0 | 82.7 | 80.7 |
| | **TOD** | **84.4** | **85.4** | **85.8** | **85.2** |

Table 3: Debiasing multiple bias attributes. We report the worst-group accuracy (%) of each bias attributes under multiple bias debiasing tasks and the average across all bias attributes. The best result is showed in **bolded**. TOD achieved the best performance under different bias settings.

2022) do not require group labels in training but still use validation sets with group labels for model selection, while methods without image supervision such as Orth-Cali require knowledge of bias attributes. Here, we consider a stricter constraint: bias information is not available in the training and validation set. Specifically, aside from the target attribute (blonde/non-blonde), we randomly select another attribute as the auxiliary prediction target and generate a dataset balanced for both targets during training. Without group information, we select the stop epoch and hyperparameters based on the average accuracy on the validation set. We compare the method with FairCLIP designed for bias-unknown scenarios. According to the results in Table 4, our method demonstrates significant improvement compared to zero-shot

| Method | WG | Avg | Gap |
|---|---|---|---|
| zero-shot CLIP | 75.9 | 82.3 | 6.4 |
| FairerCLIP | 80.4 | 84.7 | 4.3 |
| *Auxiliary Attribute* | | | |
| Heavy_Makeup | 80.7 | 84.7 | 4.0 |
| Chubby | 83.9 | 86.7 | 2.7 |
| Age | 84.4 | 88.7 | 4.2 |
| Big_nose | 85.0 | 88.5 | 3.5 |
| Gender | 86.7 | 88.4 | 1.7 |
| **Average** | **84.1** | **87.4** | **3.2** |

Table 4: Debiasing when the bias is unknown on CLIP ResNet-50.

CLIP, with an improvement of 8.2 and 5.1 on the WG and Avg, and a decrease in Gap of 3.2. It outperformed FairCLIP on all the metrics when using any of the auxiliary attribute.

## 5.4 ANALYSIS

**Ablation of Method Components.** We perform ablation study to investigate the effectiveness of the individual components of our method, with the results shown in Table 5. Both working with multi-target prediction (MTP) alone and text-only training are effective in improving group robustness. Combining the two yields further significant improvements.

In addition, Figure 2 illustrates the loss curves for the text training set and image test set in single-target and multi-target training. Because label names appear explicitly in the texts, the prompt classifier easily learns the obvious supervision signal, leading to overfitting to the text modality. As shown in Figure 2, with single-target prediction (STP) in text-only training, both the Waterbirds and CelebA datasets display a scenario where the test loss increases while the training loss decreases. This indicates that the classifier is overfitting to the text modality and fails to generalize well to the image modality. When using multi-target prediction, which is a more difficult training task, the training loss and testing loss show more consistent trends, effectively mitigating the overfitting of text-only training.

| Backbone | | | CLIP ViT-L/14 | | | | | | | | CLIP ResNet-50 | | | | |
|---|---|---|---|---|---|---|---|---|---|---|---|---|---|---|---|
| Dataset | | | **Waterbird** | | | **CelebA** | | | **Waterbird** | | | **CelebA** | | |
| Multi-Target Prediction | Text-Only Traing | WG (↑) | Avg (↑) | Gap (↓) | WG (↑) | Avg (↑) | Gap (↓) | WG (↑) | Avg (↑) | Gap (↓) | WG (↑) | Avg (↑) | Gap (↓) |
| | | 45.3 | 84.4 | 39.1 | 72.8 | 87.6 | 14.9 | 39.6 | 77.3 | 37.7 | 75.9 | 82.3 | 6.4 |
| √ | | 47.2 | 87.2 | 40.0 | 78.4 | 81.4 | 3.0 | 44.4 | 84.3 | 39.9 | 82.6 | 86.3 | 3.7 |
| | √ | 71.5 | 81.9 | 10.4 | 80.0 | 84.8 | 4.8 | 71.5 | 81.9 | 10.4 | 85.0 | 87.3 | 2.3 |
| √ | √ | **87.8** | 88.8 | **1.0** | **85.3** | 86.5 | **1.2** | **84.0** | 85.3 | **1.3** | **86.4** | 88.3 | **1.8** |

Table 5: Abaltion results on ours text-only training and multi-target prediction scheme.

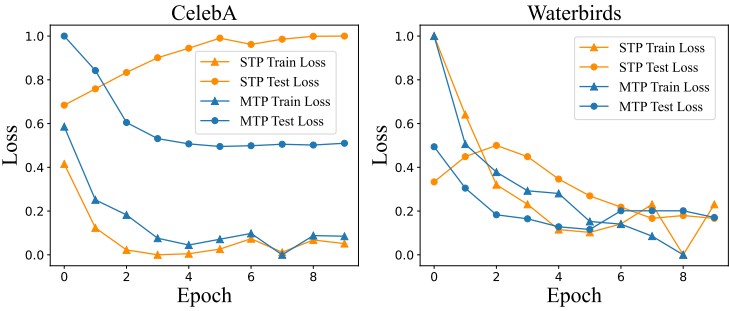

Figure 2: The loss curves on Waterbirds and CelebA dataset. The orange and blue lines represent single-target and multi-target training, respectively. Triangle marks denote training loss, and circle marks denote testing loss. We present the normalized loss curves to eliminate the dimensional impact of losses under different prediction targets.

**Attention Visualization.** To further analyze how TOD mitigates model bias, we visualize the attention distribution on images for zero-shot CLIP and TOD. We use Grad-CAM to generate the attention map of the target attribute and bias attribute based on the cosine similarity between corresponding local text embeddings and global image embeddings. As shown in Figure 4, the zero-shot CLIP leads to a global attention focus, whereas in multi-task prediction, the target attribute token focuses on the bird area while the bias attribute token concentrates on the background area. This demonstrates that the model trained on MTP can distinguish between target information and bias information, allowing the two objects to respectively focus on the corresponding image regions and thus avoiding reliance on bias attributes for target class prediction.

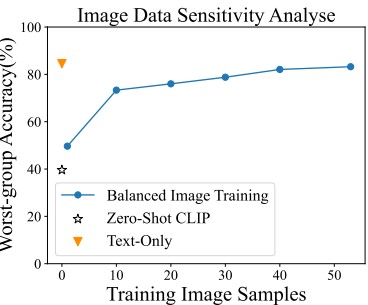

Figure 3: Image data sensitivity analyse.

| Label | Input Image | ZS CLIP | TOD (Target) | TOD (Bias) |
|-------|-------------|---------|--------------|------------|
| Landbird waterscape | | | | |
| Landbird landscape | | | | |

Figure 4: Grad-CAM (Selvaraju et al., 2017) to visualize the effect of zero-shot (ZS) CLIP and TOD. We get the embedding feature of label or bias attribute names in the text, and the highlighted areas indicate the attention of the token embedding to the image.

**Comparison of Text-based and Image-based Training.** To show to what extent our method can replace image samples, we perform multi-objective prediction training on an image dataset. We draw equal amount of image samples for each group from the Waterbirds training set to construct a balanced training set for multi-target training. Figure 3 summarizes the worst group accuracies for different amounts of samples in the training set. The one-shot training gives a small improvement in group robustness, while the worst group accuracy is still slightly lower than text-only training by 1.3 pp when using the full image training set (53 samples per group). This exhibits the sensitivity of the debiasing effect to the size of the image training set. Consequently, in scenarios where acquiring an adequate amount of image data is challenging, text-only training emerges as a viable and effective alternative.

## 6 CONCLUSION

In this paper, we propose a text-only debiasing method for CLIP. We perform prompt tuning on a balanced text dataset generated by a large language model. Further, we observe that text-only training may lead to overfitting. To address this issue, we propose multi-target prediction that predicts both target and bias attributes. Experiments show that our method achieves comparable performance to image-supervised methods and can seamlessly extend to multi-bias and bias-unknown scenarios. Future work may apply text-only training to address bias issues in other vision tasks, such as image-text retrieval.

### AUTHOR CONTRIBUTIONS

If you'd like to, you may include a section for author contributions as is done in many journals. This is optional and at the discretion of the authors.

### ACKNOWLEDGMENTS

Use unnumbered third level headings for the acknowledgments. All acknowledgments, including those to funding agencies, go at the end of the paper.

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

## A APPENDIX

### A.1 EXPERIMENTAL SETTINGS

For all methods evaluated in our experiments, including both baselines and our approach, We initialize the prompt as "This is a picture of a" for Waterbirds and "A photo of a people with" for CelebA. We employ an SGD optimizer with weight decay set to 5e-4 and a momentum of 0.9. The models are trained for 10 epochs for each dataset with the batch size of 256. We use one epoch to warmup. The learning rate and warmup rate for each dataset and model architect architecture are shown in Table 6.

| Backbone | CLIP ViT-L/14 | | CLIP ResNet-50 | |
|---|---|---|---|---|
| Parameter | **Waterbird** | **CelebA** | **Waterbird** | **CelebA** |
| batch size | 256 | 256 | 256 | 256 |
| total epoch | 10 | 10 | 10 | 10 |
| optimizer | SGD | SGD | SGD | SGD |
| lr | 1.5e-4 | 1e-5 | 1.03e-4 | 5e-6 |
| warmup lr | 1.5e-4 | 1e-5 | 1.03e-4 | 5e-6 |
| warmup epoch | 1 | 1 | 1 | 1 |
| momentum | 0.9 | 0.9 | 0.9 | 0.9 |
| weight decay | 5e-4 | 5e-4 | 5e-4 | 5e-4 |
| logit scale | 4 | 4 | 4 | 8 |
| initialization | "This is a picture of a" | "A photo of a people with" | This is a picture of a | "A photo of a people with" |

Table 6: Experimental Settings

## A.2 Text Dataset Construction for CelebA

**Attribute Description Generation.** Firstly, we utilize GPT-4o to determine contained within the dataset. The input instruction is "What attributes can be used to describe a human face?". Then, we We applied the GPT-4o to generate descriptions for each attributes. The instruction is "Please generate a collection of short descriptions of various attribute name for me.". The resulting attribute set and the amount of descriptions for each attribute is: {hair color:2, gender:2, age:4, expression: 30, beard:9, eyebrows:10, hairstyle:34, mouth:9, eyes:53, nose:13 , skin:17, face shape:13, body shape:9, decoration:6}."

**Attribute Combination with Automatic Annotation.** We compose the attributes to construct a complete image description. First, we randomly select an attribute description from the target attribute (hair color) set and the bias attribute (gender or other) set, respectively, and descriptions of 3 attributes from the other attribute set. Then, we concatenate these attribute descriptions in a random order to generate a complete textual description of a human face, and automatically obtain attribute labels. To ensure a balanced training dataset, we generated 2500 descriptions for each target-bias attribute group.

