# OpenReview forum: "Debiasing Vison-Language Models with Text-Only Training"
_ICLR.cc/2025/Conference — ICLR 2025 Conference Withdrawn Submission_

### Official Review · Reviewer_dNdD · 2024-10-31

**Soundness:** 3
**Presentation:** 2
**Contribution:** 2
**Rating:** 3
**Confidence:** 3

**Summary:**

This paper proposes a text-only debiasing method for CLIP. To address the problem that text-only training may lead to overfitting, a multi-target prediction strategy is also proposed. Extensive experiments on Waterbirds and CelebA benchmarks are conducted to validate the effectiveness of the proposed method.

**Strengths:**

1. Reducing bias in vision-language models through text-only training is an interesting topic.
2. The experimental results on Waterbirds and CelebA achieving performance comparable to SOTA image-supervised methods.
3.  Figure 2 visually demonstrates the motivation of Multi-Target Prediction.

**Weaknesses:**

1. Not ready for submission:  less than 10 pages.
2. The writing is poor, many typos, like 'use using' in Line 186, 'sSO' in Line278.
3. The text generation and MTP are limited in novelty, although  the effect seems to be okay.

**Questions:**

1. Where can the Multi-Target-Multi-Target Prediction be reflected in Figure 1?
2. The authors are suggested to supplement the changes of Grad-CAM when Text-Only Training and Multi-Target Predictionare introduced.
3. Can the method be used to improve the accuracy of zero-shot classification tasks, such as ImageNet.

---

### Official Review · Reviewer_6nFZ · 2024-11-02

**Soundness:** 3
**Presentation:** 2
**Contribution:** 2
**Rating:** 5
**Confidence:** 3

**Summary:**

This paper proposes a Text-Only Debiasing (TOD) framework for the bias problem in CLIP, in which a text-as-image training paradigm is leveraged to mitigate visual biases since texts and images are well aligned in the CLIP embedding space. For this purpose, the authors utilize a large language model to generate a balanced text dataset from the target classes and bias attributes, and introduce a Multi-Target Prediction task to mitigate the overfitting cased by the leakage of textual supervision signals. Experimental results on the Waterbirds and CelebA datasets showcase the effectiveness of the TOD framework in mitigating CLIP's bias issue, while reasonable ablation studies confirm the importance of its key components.

**Strengths:**

- This paper is generally well organized and presented.
- It is well motivated to take text as image in CLIP embedding space and generate balanced text data to address the bias issue of CLIP.
- Overall, the experiments in the paper are quite thorough, especially the loss curve in Figure 2, which effectively validates the effectiveness of the multi-objective prediction task in alleviating overfitting.

**Weaknesses:**

1. The paper needs careful proofreading, some writing errors are as follows:
- demonstrate -> demonstrates, line 041;
- false attributes -> bias attributes, line 183;
- Both training and inference process is -> Both training and inference processes are, line 184;
- we use using -> we use, line 183;
- $C_b$ -> $C_B$, line 244;
- $\frac{<\cdot,\cdot>\tau}{\cdots}$ -> $\frac{<\cdot,\cdot>/\tau}{\cdots}$, Eq.2 and Eq.4.
- sSo -> So, line 277;
2. Eq.5 is confusing. If I understand correctly, $p(y=i,b=j|x)$ is a joint probability of the target class and bias attribute, and how to compute $\mathrm{max}_j\ p(y=i,b=j|x)$ firstly?
3. Given that the training of the TOD framework necessitates executing the forward process of the CLIP text encoder for both the prompts and input text, the authors are encouraged to evaluate its training efficiency (time per step, GPU memory usage) in comparison to baseline models.
4. The contribution of this paper is somewhat limited. As the core contribution of the paper, the text-as-image training paradigm has been previously proposed in earlier works [1, 2, 3]. Besides, LLM-based instruction-following text generation from categories is also not fresh [2, 4]. To highlight the contribution of this paper, the authors are advised to explore different instruction templates and more efficient ways of generating text in terms of debiasing the CLIP model. Furthermore, although CLIP has well aligned texts and images in a unified embedding space, the modality gap between them still objectively exists. Therefore, finding ways to overcome this modal gap and enhance the cross-modal transfer ability of the TOD model will make this work more solid.

[1] Texts as Images in Prompt Tuning for Multi-Label Image Recognition, CVPR 2023.

[2] Text as Image: Learning Transferable Adapter for Multi-Label Classification, arXiv 2023.

[3] TAI++: Text as Image for Multi-Label Image Classification by Co-Learning Transferable Prompt, IJCAI 2024.

[4] Language-Driven Cross-Modal Classifier for Zero-Shot Multi-Label Image Recognition, ICML 2024.

**Questions:**

The overfitting issue caused by the leakage of textual supervision signals is also identified by the paper [1], in which random perturbation is proposed to mitigate overfitting by perturbing text embeddings with noise. Compared to the multi-task prediction in this paper, which strategy performs better in dealing with the overfitting problem?

[1] Text as Image: Learning Transferable Adapter for Multi-Label Classification, arXiv 2023.

---

### Official Review · Reviewer_g5RA · 2024-11-03

**Soundness:** 3
**Presentation:** 2
**Contribution:** 2
**Rating:** 5
**Confidence:** 4

**Summary:**

This paper proposes a text-only debiasing method for the vision-language model debiasing task. The method first uses an LLM to generate an attribute balanced Dataset, followed by prompt tuning. Through multi-target prediction, it simultaneously predicts target attributes and bias attributes. Experimental results demonstrate the effectiveness of this approach, as well as its handling of multiple bias attributes and unknown bias cases.

**Strengths:**

- The authors use the design of the Attribute Balanced Dataset to achieve debiasing with only text information.

- The experiments are thorough, analyzing not only standard benchmarks but also cases with multiple bias attributes and unknown bias.

**Weaknesses:**

- This paper claims to debias VLMs, but only evaluates on CLIP models in the experiments. Can the proposed method be applied to debias other VLMs as well?

- The proposed method involves prompt tuning, which could be costly. Could the authors provide a detailed time comparison with other baselines?

**Questions:**

Please refer to the weakness above. I will carefully review the rebuttal and consider the opinions of the other reviewers to adjust my rating.

---

### Official Review · Reviewer_1JsM · 2024-11-05

**Soundness:** 3
**Presentation:** 3
**Contribution:** 3
**Rating:** 5
**Confidence:** 4

**Summary:**

The paper addresses debiasing in the context of Vision-Language Models (VLMs). Specifically, the authors argue that existing methods for debiasing VLMs struggle to obtain sufficient images to represent minority groups, along with high costs for labeling such minority groups. To mitigate these issues, the authors propose Text-Only Debiasing (TOD) - a simple prompt-tuning-based framework to debias VLMs through text-only training. TOD generates a balanced text-only dataset using GPT-4 and performs prompt tuning using the same. However, this faces the potential issue of the model overfitting to the text modality. To overcome this, the authors propose a Multi-Target Prediction (MTP) task to predict the index of the target and bias attributes. Experiments on the CelebA and Waterbirds datasets demonstrate the effectiveness of the proposed approach.

**Strengths:**

- **Motivation**—The authors propose a text-only framework to mitigate the expense of image-based fine-tuning. This is based on the fact that limited data is available for minority groups, and labeling them can be quite expensive. Through a text-only approach utilizing LLMs, the proposed method can circumvent this expense and generate a balanced dataset for prompt tuning.
- **Results** - The proposed method TOD achieves significant improvements over prior works in various attribute settings in CelebA and Waterbird, demonstrating the effectiveness of the approach.

**Weaknesses:**

### (a) Intuition behind the proposed approach
- The proposed approach - Text-only Debiasing (TOD), is based on the idea that the optimized learnable text prompts are directly applicable to the image modality due to the image and text modalities sharing a unified representation space. However, [R1] shows that the image and text embeddings of CLIP are located in two separate regions in the representation space.
- [R1] essentially contradicts one of the fundamental motivating ideas for TOD. This can imply one of the following - (i) TOD is impervious to the modality gap in CLIP or (ii) TOD somehow implicitly bridges the modality gap, which seems unlikely, based on the reviewer’s understanding since there is no training for the image encoder. Can the authors discuss the proposed method in the context of this paper and explain how TOD works despite this modality gap in CLIP?
- Additionally, since the whole proposed framework hinges on this property of CLIP, can the authors present simple motivating zero-shot experiments to show that the optimized prompts are directly applicable to the image encoder?

### (b) Experiments and results
- **Choice of bias attributes-** In L373-375, the authors list gender bias, age bias, and wavy hair bias. Similarly, in L353-355, they consider chubby, wearing a hat, and age as various bias attributes. However, there is no discussion on these choices of bias attributes from CelebA. Is there a specific reason behind the choice of these attributes, or were they randomly chosen? The authors should consider presenting a comprehensive list of experiments in the supplementary on various bias attributes of CelebA to demonstrate the effectiveness of the approach.
- **Unknown bias attributes-** In L416-422, the authors claim that existing works such as Orth-Cali require knowledge of bias attributes, while TOD does not. However, the reviewer feels that the description of TOD does not reflect this. In this experiment, the authors select an attribute at random to serve as an auxiliary attribute based on which a balanced dataset is generated. Doesn’t this count as “requiring knowledge of bias attributes” similar to Orth-Cali? Orth-Proj / Orth-Cali use knowledge of the bias attribute to obtain a projected representation without the bias attribute, while TOD uses knowledge of the bias attribute to generate a balanced dataset. Essentially, both of these methods use information of the bias attribute in some way. Could the authors clarify how TOD does not require knowledge of the bias attribute, despite the discussion seemingly pointing the other way?

- **Results in Fig. 4-** The authors demonstrate in Fig.4 that text-only training can perform on par with image-based training. However, there is no clear explanation of the experimental setup of the balanced image training setting. Can the authors provide some discussion on the details of this setup?
### (c) Minor writing issues
- There are formatting errors (Eg. Author contributions and Acknowledgments left unchanged from the template) and grammatical errors (Eg: L704-707) in the paper. Additionally, there is an error in the heading of Table 1, i.e., the top section of Table 1 should be “methods with image data”. The reviewer suggests that the authors go through the entire paper to rectify such issues.

### (c) Missing references
- R1 - Liang, Victor Weixin, et al. "Mind the gap: Understanding the modality gap in multi-modal contrastive representation learning." NeurIPS 2022..
- R2 - Seth, Ashish, Mayur Hemani, and Chirag Agarwal. "Dear: Debiasing vision-language models with additive residuals." CVPR 2023.

**Questions:**

- Following the discussion from the second point under (b) in Weaknesses, can the authors clarify how TOD deviates from Orth-Cali and does not require prior knowledge of the bias attribute, despite the explanation indicating otherwise?
- How was the balanced image training experiment in Fig. 4 done? Can the authors provide further details of this experiment?
- Can the authors present results on the FairFace dataset in addition to CelebA and Waterbirds? It would make the experiments comprehensive and more complete.

---

### Note · Authors · 2024-11-13

I have read and agree with the venue's withdrawal policy on behalf of myself and my co-authors.